# Regorafenib Treatment for Recurrent Glioblastoma Beyond Bevacizumab-Based Therapy: A Large, Multicenter, Real-Life Study

**DOI:** 10.3390/cancers17010046

**Published:** 2024-12-27

**Authors:** Salih Tünbekici, Haydar cagatay Yuksel, Caner Acar, Gökhan Sahin, Seval Orman, Nargiz Majidova, Alper Coskun, Mustafa Seyyar, Mehmet sıddık Dilek, Mahmut Kara, Ahmet Kursat Dıslı, Teyfik Demir, Nagihan Kolkıran, Mustafa Sahbazlar, Erkut Demırcıler, Fatih Kuş, Ali Aytac, Serkan Menekse, Hakan Yucel, Sedat Biter, Tolga Koseci, Ahmet Unsal, Ahmet Ozveren, Alper Sevınc, Erdem Goker, Pınar Gürsoy

**Affiliations:** 1Department of Medical Oncology, Ege University Medical Faculty, Izmir 35040, Turkey; haydar.cagatay.yuksel@ege.edu.tr (H.c.Y.); caner.acar@ege.edu.tr (C.A.); gokhan.sahin@ege.edu.tr (G.S.); pinargursoy77@gmail.com (P.G.); 2Department of Medical Oncology, Kartal Dr. Lütfi Kirdar City Hospital, Health Science University, Cevizli, D-100 Güney Yanyol, Cevizli Mevkii No:47, Kartal, Istanbul 34865, Turkey; seval.orman@saglik.gov.tr; 3Department of Medical Oncology, School of Medicine, Marmara University, Istanbul 34899, Turkey; nergiz.mecidova1991@gmail.com; 4Department of Medical Oncology, Uludağ University, Bursa 16059, Turkey; alpercoskun@uludag.edu.tr; 5Department of Medical Oncology, Gaziantep City Hospital, Gaziantep 27470, Turkey; mustafaseyyar27@hotmail.com; 6Medical Oncology, Medical School, Dicle University, Diyarbakir 21280, Turkey; msiddik.dilek@dicle.edu.tr; 7Department of Medical Oncology, Yuzuncu Yil University Faculty of Medicine, Van 65090, Turkey; mkara@yyu.edu.tr; 8Department of Medical Oncology, Erciyes University Faculty of Medicine, Kayseri 38030, Turkey; ahmetkursat@erciyes.edu.tr; 9Department of Medical Oncology, Ondokuz Mayis University Faculty of Medicine, Samsun 55270, Turkey; teyfik.demir@omu.edu.tr; 10Department of Medical Oncology, Celal Bayar University Faculty of Medicine, Manisa 45030, Turkey; nagihan.kolkiran@cbu.edu.tr (N.K.); mustafa.sahbazlar@cbu.edu.tr (M.S.); 11Department of Medical Oncology, 9 Eylül University Faculty of Medicine, Izmir 35220, Turkey; erkut.demirciler@deu.edu.tr; 12Department of Medical Oncology, Hacettepe University Cancer Institute, Ankara 06230, Turkey; fatihkus@hacettepe.edu.tr; 13Department of Medical Oncology, Mehmet Akif İnan Training and Research Hospital, Sanlıurfa 63040, Turkey; dr_aliaytac@hotmail.com; 14Department of Medical Oncology, Manisa City Hospital, Manisa 45040, Turkey; drsermen@hotmail.com; 15Department of Medical Oncology, School of Medicine, Gaziantep University, Gaziantep 27580, Turkey; drhakan88@gmail.com; 16Department of Medical Oncology, Cukurova University, Adana 01790, Turkey; sedatb23@hotmail.com (S.B.); tkoseci@cu.edu.tr (T.K.); 17Department of Medical Oncology, Gumushane State Hospital, Gumushane 29000, Turkey; ahmetnsal@gmail.com; 18Medical Oncology, Department MD, İzmir Kent Hospital, Izmir 35620, Turkey; ahmet.ozveren@acibadem.com; 19Medical Oncology, Medical Park Gaziantep Hospital, Gaziantep 27090, Turkey; alper.sevinc@mph.com.tr

**Keywords:** regorafenib, recurrent glioblastoma, safety, efficacy, targeted therapy, real-world

## Abstract

Despite recent advances, glioblastoma remains incurable, with a poor prognosis. The REGOMA trial compared regorafenib with lomustine as second-line treatments for recurrent glioblastoma. Regorafenib resulted in better overall survival than lomustine, leading to its inclusion in current guidelines. Currently, bevacizumab, lomustine, and regorafenib are recommended treatment options for recurrent glioblastoma. Bevacizumab is a vascular endothelial growth factor receptor-A inhibitor and prevents angiogenesis. However, the optimal treatment for patients whose conditions progress after bevacizumab-based therapy in the third-line setting remains unclear. We assessed the efficacy and safety of regorafenib as a third-line treatment after bevacizumab and aimed to evaluate the potential benefit of continued vascular endothelial growth factor receptor inhibition. Regorafenib appears to be a good option regarding efficacy and safety for patients who progress after bevacizumab-based therapy. However, further studies are required to better define the role of regorafenib in recurrent glioblastoma.

## 1. Introduction

Glioblastoma is the most common malignant primary brain tumor in adults [1]. The high risk of recurrence or progression in glioblastoma, along with its limited treatment options, leads to a poor prognosis. Relative survival rates for glioblastoma are notably low, with only 6.8% of patients surviving 5 years after their diagnosis [2,3].

The standard treatment for newly diagnosed glioblastoma includes maximal surgical resection followed by radiotherapy with concurrent temozolomide and then adjuvant temozolomide [4,5]. A study showed that approximately 72% of patients with glioblastoma relapsed within 1 year and that approximately 86% of patients relapsed within 2 years [6]. In cases of recurrence, resurgery and re-irradiation may be considered for suitable patients, but systemic therapy remains the most commonly used approach [7].

Glioblastoma is one of the most vascularized solid tumors, with vascular proliferation being a key pathological feature [8]. In glioblastoma, the blood vessels typically consist of abnormal glomeruloid vascular structures, formed by intricate clusters of newly developed microchannels. These channels are lined with hyperplastic endothelial cells that exhibit altered morphological features and are supported by the basal lamina and pericytes. These blood vessels are structurally and functionally aberrant, contributing to the creation of a hostile microenvironment. The presence of these abnormal vessels is a key histopathological characteristic of glioblastoma [9].

Angiogenesis, a crucial factor in tumor progression, is regulated by interconnected signaling pathways. Vascular endothelial growth factor (VEGF), the first vascular-specific growth factor identified, plays a pivotal role in driving tumor angiogenesis. The three closely related members of the vascular endothelial growth factor receptor (VEGFR) family include VEGFR2, which plays a significant role in mediating the primary effects of VEGF on blood vessel growth and permeability. Beyond VEGF and its receptors, numerous other growth factors and receptors function synergistically through established pathways to regulate both tumor angiogenesis and growth.

These pathways include fibroblast growth factor receptor, activated by a specific set of ligands that drive tumor cell proliferation and differentiation via various downstream signaling cascades, as well as platelet-derived growth factor receptor, which aids in vessel stabilization by modulating pericyte recruitment and maturation. Tyrosine kinase with immunoglobulin and epidermal growth factor homology domain 2 (TIE2) serves as a key regulator of angiogenesis. Predominantly or exclusively expressed in endothelial cells, TIE2 is crucial for the maturation of immature vessels through interactions with angiopoietin 1, angiopoietin 2, VEGF, and fibroblast growth factor ligands. Both VEGFR2 and TIE2 play essential roles in the function of normal and tumor vasculature [10].

Current guidelines recommend nitrosoureas, bevacizumab, and regorafenib as second-line treatments after progression following first-line therapy [11]. Lomustine has been used in many studies as a second-line treatment [12,13,14,15,16]. Bevacizumab is a recombinant humanized antibody that inhibits VEGFR-A and prevents angiogenesis. Bevacizumab has been studied in various trials of newly diagnosed and recurrent glioblastoma [17,18,19,20,21].

Regorafenib is an oral multikinase inhibitor that targets kinases involved in tumor angiogenesis, oncogenesis, and the maintenance of the tumor microenvironment, resulting in the inhibition of tumor growth. Regorafenib shows anti-angiogenic activity owing to its dual-targeted VEGFR-TIE2 tyrosine kinase inhibition [22,23]. Additionally, by inhibiting molecular escape pathways to VEGF inhibition, regorafenib continues to demonstrate anti-angiogenic effects, even in tumors that have developed resistance to VEGF inhibitors. Preclinical and clinical studies have provided evidence supporting regorafenib’s continuous anti-angiogenic effects in tumors resistant to other VEGF inhibitors, as well as its role in remodeling the tumor microenvironment [24]. Regorafenib is currently approved for use in colorectal cancer, hepatocellular carcinoma, and gastrointestinal stromal tumors [25,26,27]. Regorafenib has demonstrated antitumor effects in multiple cell lines, including gliomas. The antitumor activity of regorafenib is associated with a reduction in tumor vascularization and the induction of apoptotic cell death [28]. The results of the REGOMA trial, which tested regorafenib compared with lomustine in patients with recurrent glioblastoma, were reported in 2019 [29]. This trial showed that overall survival (OS) in the regorafenib arm was better than that in the lomustine arm. Based on these results, regorafenib was included in the guidelines as the preferred regimen for patients with recurrent glioblastoma.

After progression following first-line therapy, the optimal treatment option for patients with glioblastoma remains unclear. In the current clinical guidelines, nitrosoureas, bevacizumab, and regorafenib are recommended as second-line treatments [11]. However, in Turkey, access to nitrosoureas is limited, and because regorafenib is reimbursed only as a third-line treatment, bevacizumab-based therapies are generally preferred for second-line treatment. Therefore, this study aimed to evaluate the efficacy and safety of regorafenib in patients with bevacizumab-refractory glioblastoma in a real-life setting and to investigate the potential benefits of continuous VEGFR inhibition.

## 2. Materials and Methods

### 2.1. Study Design and Participants

This retrospective, multicenter study, which was based on a review of clinical records, included patients treated between January 2021 and December 2023 across 19 oncology centers in Turkey. We investigated the role of regorafenib in patients with bevacizumab-refractory glioblastoma. The inclusion criteria were as follows: histologically confirmed diagnosis of isocitrate dehydrogenase (IDH)-wildtype glioblastoma, progression after second-line bevacizumab-based treatment, age >18 years, Eastern Cooperative Oncology Group (ECOG) performance status score ≤2, and adequate bone marrow, liver, and renal function. The exclusion criteria were as follows: IDH-mutant high-grade gliomas; uncontrolled hypertension; active or chronic hepatitis B or C virus infection requiring antiviral treatment; prior treatment with regorafenib; arterial thrombotic or embolic events, pulmonary embolism, or myocardial infarction occurring within 6 months before regorafenib therapy; and the use of strong cytochrome P450 3A4 inhibitors or inducers.

The patients’ clinical, pathological, and radiological data were recorded from the patients’ files. Regorafenib was administered orally at a dose of 160 mg (four 40 mg tablets) daily for 3 weeks within a 4-week cycle. Dose reductions were implemented on the basis of the severity of toxicity, with a minimum daily dose of 80 mg. Clinical follow-up was performed every 2 weeks, and gadolinium-enhanced brain magnetic resonance imaging was performed every 8 weeks to evaluate the response to treatment. Responses were evaluated using the Response Assessment in Neuro-Oncology criteria. Adverse events were classified and graded according to the National Cancer Institute’s Common Terminology Criteria for Adverse Events version 5.0.

### 2.2. Statistical Analysis

Categorical variables are shown as the frequency (n) and percentage (%), while numerical data are presented as the median and range to represent the distribution. Progression-free survival (PFS) was defined as the time from the start date of regorafenib to the date of progression or death. OS was defined as the time from initiating regorafenib to death. The objective response rate was defined as the percentage of patients with a complete response (CR) + partial response (PR) according to the Response Assessment in Neuro-Oncology criteria. The disease control rate (DCR) was defined as the percentage of patients with a CR + a PR + stable disease (SD). The OS and the PFS were estimated using the Kaplan–Meier method. Regarding OS, patients who were still alive were censored from the date of analysis. Patients without progression were censored at their final follow-up visit. The log-rank test was used for a univariate analysis. Variables with *p* < 0.05 were considered statistically significant. All analyses were performed using IBM SPSS Statistics software (version 26; IBM Corp., Armonk, NY, USA).

## 3. Results

### 3.1. Patient Characteristics

The general characteristics of the patients with recurrent glioblastoma are detailed in Table 1. A total of 65 patients were included in this study. The median age of the patients was 53 (18–67) years. Thirty-nine (60%) patients were men and twenty-six (40%) were women. While the tumor was completely resected at the first presentation in 41 (63.1%) patients, it was only partially resected in 24 (36.9%) patients. Corticosteroid use during regorafenib treatment was reported in 35 (53.8%) patients. After progression, 12 (18.5%) patients received further treatment, including various chemotherapeutic regimens, while 53 (81.5%) patients did not receive additional treatment. A methylated methylguanine–DNA methyltransferase (MGMT) test result was obtained from seven patients, one of whom was methylated and six were not. The MGMT status of 58 patients remained unknown.

### 3.2. Survival Outcomes

At the time of the analysis, 61 (93.8%) patients had died. The median progression-free survival (mPFS) and median overall survival (mOS) were 2.5 months (95% Confidence Interval: 2.23–2.75) and 4.1 months (95% Confidence İnterval: 3.52–4.68), respectively (see Figure 1 and Figure 2). In the univariate analysis, age, sex, steroid use, secondary surgery, treatment after regorafenib, and the ECOG performance status were evaluated for their association with PFS and OS. The univariate analysis revealed that patients who received subsequent therapy after regorafenib had a significantly higher OS benefit (5.13 months 95% CI: 4.04–6.27) compared to those who did not (3.91 months 95% CI: 3.41–4.41) (*p* = 0.022). The PFS was longer in patients with ECOG 0–1 (2.69 months 95% CI: 2.19–3.19) than in those with ECOG 2 (2.10 months 95% CI: 1.67–2.53) (*p* = 0.042) (see Table 2).

### 3.3. Response to Regorafenib

The DCR was 23%. An objective response was observed in three (4.6%) patients, with no CR recorded. A PR was observed in 3 (4.6%) patients, SD in 12 (18.4%), and progressive disease in 36 (55.3%). The treatment response could not be evaluated in 14 (21.5%) patients; 10 of these patients died, and 4 discontinued regorafenib because of adverse effects (see Table 3).

### 3.4. Safety

Eleven (16.9%) patients required a dose reduction owing to drug-related adverse events. At least one drug-related side effect was observed in 55 (84.6%) patients, with the majority classified as Grade 1 or 2. Twenty (30.8%) patients experienced Grade 3–4 adverse effects. The most commonly observed Grade 3–4 toxicities included hand–foot skin reactions (6 patients), elevated bilirubin concentrations (2 patients), hypertransaminasemia (2 patients), skin rash (2 patients), and fatigue (3 patients). Four (6.2%) patients permanently discontinued treatment because of drug-related side effects (one because of thrombocytopenia, one because of pulmonary embolism, one because of elevated lipase, and one because of fatigue). No deaths were considered drug-related (see Table 4).

## 4. Discussion

To the best of our knowledge, this is the first study to evaluate the efficacy and safety of continuous VEGFR inhibition and regorafenib treatment in real-life patients with glioblastoma who progressed after bevacizumab-based therapy. We anticipate that our findings will contribute to the literature because of the lack of real-life data for third-line treatment with regorafenib.

The REGOMA trial included patients with glioblastoma who progressed after temozolomide treatment and compared regorafenib with lomustine in a second-line setting [29]. In the regorafenib arm, the mPFS was 2.0 months, the mOS was 7.4 months, and the DCR was 44%. In 2021, Lombardi et al. published real-life retrospective data on the effectiveness of regorafenib in recurrent glioblastoma [30]. This study included 54 patients; the mPFS was 2.3 months, the mOS was 10.2 months, and the DCR was 46.3%. Although the OS was slightly higher, the results were generally similar to those of the REGOMA trial. In our study, the efficacy and safety of regorafenib were evaluated in a third-line setting for patients who progressed after bevacizumab. Our study showed an mPFS of 2.5 months, an mOS of 4.1 months, and a DCR of 23%. We consider that the lower mOS rate in our study is due to regorafenib being used in a later line of treatment and 30% of our patients having an ECOG performance status of 2.

Regarding safety, in the REGOMA trial, 33 (56%) patients experienced Grade 3–4 adverse events, 10 (17%) required dose reduction because of adverse events, and 4 (7%) had to permanently discontinue the drug because of adverse events [29]. In our study, when we analyzed the safety of regorafenib in the third-line setting, we found that four (6.9%) patients required permanent discontinuation of the drug owing to adverse events. Additionally, 11 (16.9%) patients required a reduction in dose owing to adverse events. These findings are similar to those of the REGOMA trial. However, the number of patients who developed Grade 3–4 adverse events in our study was 20 (30.7%), which is lower than that in the REGOMA trial. This analysis demonstrates that regorafenib can be used safely in a third-line setting.

Current guidelines recommend bevacizumab, lomustine, and regorafenib as second-line treatments for patients with glioblastoma who have had their first relapse [11]. However, in patients who progress after second-line treatment, there are no reliable data available regarding the optimal therapeutic options. There is no effective treatment for patients with recurrent glioblastoma who progress while using second-line bevacizumab-based therapy. Bevacizumab is typically continued in patients who progress after bevacizumab-based treatment, with additional agents added or the concomitant agent switched. However, the expected clinical benefit may not be achieved, and the toxicity associated with the added chemotherapy agents is increased.

A study that investigated the efficacy of adding nitrosoureas to bevacizumab therapy in patients who progressed while on initial bevacizumab treatment showed that the addition of nitrosoureas did not provide any benefits and led to increased toxicity. Additionally, the ORR was 0%, and the mPFS was 6.3 weeks [31]. Quant et al. showed that, in patients who progressed after bevacizumab-based therapy, continuing bevacizumab while changing the accompanying chemotherapy regimen resulted in an mPFS of 5 weeks, and only 2% of patients were progression-free at 6 months. Most of these patients received carboplatin as a chemotherapy agent during second-line bevacizumab-based treatment [32]. Another study showed that in patients with glioblastoma who progressed after bevacizumab-based therapy, bevacizumab was continued and additional dasatinib was added. The mPFS was 4 weeks, and all patients had progressed by 6 months, with an mOS of 2.5 months [33]. Ahluwalia et al. investigated the efficacy of the endoglin inhibitor TRC105 in bevacizumab-refractory patients [34]. They found that the mPFS was 1.4 months when TRC105 was administered alone, suggesting that it had no major effect on its own. However, when TRC105 was combined with bevacizumab, the mPFS increased to 1.8 months, and the mOS was 5.7 months. In our study, we included patients with glioblastoma who progressed after bevacizumab-based therapy, and the mPFS was 2.5 months, with an mOS of 4.1 months. Furthermore, 4.6% of patients remained progression-free at 6 months. In our study, the fact that three (4.6%) patients were still progression-free at 6 months raises the possibility that regorafenib use and continued VEGFR inhibition could be effective in a subset of patients. However, further studies are needed to identify which patients these are.

As expected, there are several limitations because of the nature of this study. Potential bias may have been introduced because of our study’s retrospective design. Adverse effects may have been underreported. Additionally, because this was a multicenter study, evaluations were not performed by the same neuroradiologist. Furthermore, because the MGMT test is not routinely conducted in Turkey, the majority of patients did not have their MGMT status assessed.

## 5. Conclusions

To the best of our knowledge, this is the first study to assess the efficacy and safety of regorafenib in a third-line setting for patients with recurrent glioblastoma who progressed while on bevacizumab-based therapy in a real-life population. Additionally, we addressed the question of whether continuous VEGFR inhibition offers any benefits. Regorafenib could be considered a viable option in terms of efficacy and safety for patients who experience progression after bevacizumab-based therapy. We believe that this study provides important insights because of the lack of third-line data on regorafenib in recurrent glioblastoma in the literature. Further prospective and randomized studies are needed to confirm the role of regorafenib in glioblastoma.

## Figures and Tables

**Figure 1 cancers-17-00046-f001:**
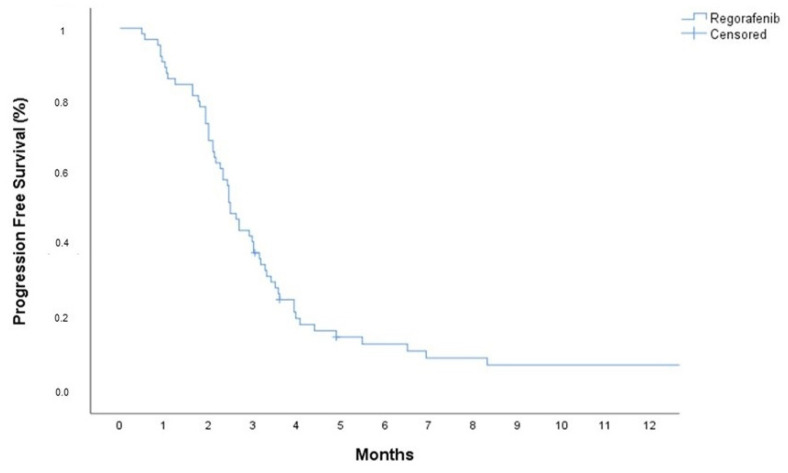
Kaplan–Meier curve of progression-free survival.

**Figure 2 cancers-17-00046-f002:**
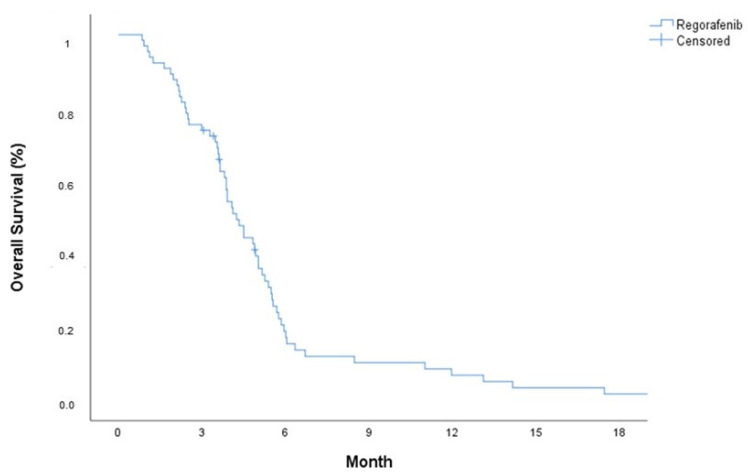
Kaplan–Meier curve of overall survival.

**Table 1 cancers-17-00046-t001:** General characteristics of patients (n = 65) with recurrent glioblastoma.

Variable	Value
Patients, n (%)	65 (100)
Age (years), median	53 (18–67)
Sex, n (%)	
Male	39 (60)
Female	26 (40)
Type of first surgery, n (%)	
Radical surgery	41 (63.1)
Partial surgery	24 (36.9)
ECOG PS, n (%)	
0	6 (9.2)
1	39 (60)
2	20 (30.8)
Corticosteroids, n (%)	
Yes	35 (53.8)
No	27 (41.5)
Missing	3 (4.6)
Second surgery, n (%)	
Yes	2 (3.1)
No	63 (96.9)
MGMT, n (%)	
Methylated	1 (1.6)
Unmethylated	6 (9.2)
Missing	58 (89.2)
IDH status, n (%)	
Wild	65(100)
Subsequent treatment after regorafenib, n (%)	12 (18.5)
Carboplatin–Paclitaxel	4 (6.1)
Bevacizumab–Temozolomide	2 (3)
Bevacizumab–İrinotecan	2 (3)
Lomustine	2 (3)
Carmustine	2 (3)
Exitus, n (%)	61 (93,8)

All data are presented as the median (range) or absolute number (%). Abbreviations: ECOG PS—Eastern Cooperative Oncology Group performance status; IDH—isocitrate dehydrogenase; MGMT—methylated methylguanine–DNA-methyltransferase.

**Table 2 cancers-17-00046-t002:** Univariate analysis for progression-free survival and overall survival.

	Progression-Free Survival		Overall Survival	
Variables	Median	*p*	Median	*p*
Age ≥65 <65	2.46 (2.36–2.56) 2.49 (2.11–2.87)	0.928	4.07 (3.30–4.82) 4.23 (3.52–4.95	0.909
Sex Female Male	2.49 (1.92–3.72) 2.46 (2.02–2.91)	0.234	4.83 (3.14–6.51) 4.10 (3.59–4.62)	0.090
Steroid Treatment Yes No	2.49 (1.54–3.46) 2.33 (1.88–2.79)	0.234	3.91 (3.25–4.56) 4.33 (3.69–4.97)	0.884
Secondary Operation Yes No	1.81 (NA) 2.46 (2.22–2.7)	0.45	2.99 (NA) 3.91 (3.51–4.30)	0.738
Subsequent Treatment After Regorafenib Yes No	2.49 (1.71–3.27) 2.46 (2.10–2.83)	0.71	5.13 (4.04–6.27) 3.91 (3.41–4.41)	**0.022 ***
ECOG-PS 0–1 2	2.69 (2.19–3.19) 2.10 (1.67–2.53)	**0.042 ***	4.5 (3.67–5.32) 3.48 (2.29–4.67)	0.085

In univariate analysis, ***** ECOG is statistically significant for PFS, while subsequent treatment after regorafenib is statistically significant for OS. Abbreviations: ECOG PS—Eastern Cooperative Oncology Group performance status.

**Table 3 cancers-17-00046-t003:** Best response during regorafenib treatment.

Outcome	No (%)
Complete response	0 (0)
Partial response	3 (4.6)
Stable disease	12 (18.4)
Progression disease	36 (55.3)
Not evaluated *	14 (21.5)

Objective response rate, 4.6%; disease control rate, 23%. * Radiological evaluation could not be performed for 10 patients due to death and for 4 patients due to discontinuation of regorafenib because of adverse effects. Neuroradiological assessment was carried out according to the Response Assessment in Neuro-Oncology criteria.

**Table 4 cancers-17-00046-t004:** Treatment-related adverse events according to the Common Terminology Criteria for Adverse Events v 5.0.

Grade 3–4 Drug-Related Adverse Events	Number
All	20
Hand–foot Syndrome	6
Fatigue	3
Skin Rash	2
Hypertransaminasemia	2
Blood Bilirubin Increased	2
Thrombocytopenia	1
Hypertension	1
Others	3

## Data Availability

Data will be available from the corresponding author upon reasonable request.

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
