# Peer review of "Regorafenib Treatment for Recurrent Glioblastoma Beyond Bevacizumab-Based Therapy: A Large, Multicenter, Real-Life Study"

_cancers, 2024, doi:10.3390/cancers17010046_

Round 1

Reviewer 1 Report

Comments and Suggestions for Authors

The manuscript is in raw condition to be published and needs serious revisions

Comments on the Quality of English Language

English language needs proof reading

Author Response

Comments 1: [The manuscript is in raw condition to be published and needs serious revisions.]

Response 1: [In the introduction of the manuscript, angiogenesis pathways, the relationship between regorafenib and angiogenesis, the continuous VEGFR inhibition of regorafenib, and its benefits have been discussed in detail. In the discussion section, relevant literature on regorafenib has been emphasized, and the potential benefits of continuous VEGFR inhibition have been thoroughly discussed. All necessary corrections and additions have been made.]

Comments 2: [English language needs proof reading.]

Response 2:[The English language has been improved. The necessary document is attached.]

We would like to sincerely thank the reviewers for their valuable contributions and insightful feedback, which greatly enhanced the quality of our manuscript. Your thoughtful comments and suggestions are deeply appreciated.

Reviewer 2 Report

Comments and Suggestions for Authors

This manuscript studies the efficacy and safety of regorafenib as a third-line treatment for patients with recurrent glioblastoma. Some issues need to be addressed as follows.

1. The abstract should be written in a paragraph without being classified into four sessions.

2. The captions for Figures 1 and 2 need to be proofread. There is one extra bracket. 

3. In Table 4, the asterisks need to be explained. 

Author Response

Comments 1: [The abstract should be written in a paragraph without being classified into four sessions.]

Response 1: [The abstract has been revised according to the guidelines provided in the authors' instructions and written as a single paragraph without being classified into four sections.]

Comments 2: [The captions for Figures 1 and 2 need to be proofread. There is one extra bracket.]

Response 2: [The captions for Figures 1 and 2 have been revised based on your suggestions.]

Comments 3: [In Table 4, the asterisks need to be explained.]

Response 3: [The asterisks in Table 4 have been explained based on your suggestions.]

We would like to sincerely thank the reviewers for their valuable contributions and insightful feedback, which greatly enhanced the quality of our manuscript. Your thoughtful comments and suggestions are deeply appreciated.

Reviewer 3 Report

Comments and Suggestions for Authors

This is a retrospective study that evaluates the safety and efficacy of regorafenib in recurrent GBM after Bevacizumab treatment. Overall, the manuscript is well organized and well presented. Here are minor comments:

1. Subtitles are recommended to be added in results section.

2. Proofreading is necessary. Some abbreviations are not consistently defined (IDH and isocitrate dehydrogenase).

Author Response

Comments 1: [Subtitles are recommended to be added in results section.]

Response 1: [The necessary structural adjustments have been made.]

Comments 2: [Proofreading is necessary. Some abbreviations are not consistently defined (IDH and isocitrate dehydrogenase).]

Response 2: [The proofreading has been done.The abbreviations have been corrected.]

We would like to sincerely thank the reviewers for their valuable contributions and insightful feedback, which greatly enhanced the quality of our manuscript. Your thoughtful comments and suggestions are deeply appreciated.

Reviewer 4 Report

Comments and Suggestions for Authors

The manuscript entitled "Regorafenib in recurrent Glioblastoma Patients Beyond Bevacizumab-Based Therapy: A Large and Multicenter Real-Life Study" by Tunbekici et al., described the efficacy and safety of Regorafenib as third-line treatment for recurrent glioblastoma after bevacizumab-based therapy. The article is well written and provides important insights in the field of oncology. 

I have some minor comments:

-I suggest to better describe the importance of VEGF and angiogenesis pathway in the introduction and Discussion sections. 

-I suggest to better emphasize the articles cited in the manuscript about regorafenib.

-I suggest to better emphasize the limitations of the study. 

Author Response

Comments 1: [I suggest to better describe the importance of VEGF and angiogenesis pathway in the introduction and Discussion sections.]

Response 1: [Detailed information has been provided regarding the angiogenesis pathways and the importance of VEGF.]

Comments 2: [I suggest to better emphasize the articles cited in the manuscript about regorafenib.]

Response 2: [The relevant emphasis has been placed on the articles about regorafenib.]

Comments 3: [I suggest to better emphasize the limitations of the study.] 

Response 3: [The necessary emphasis has been placed on the limitations of the study.]

We would like to sincerely thank the reviewers for their valuable contributions and insightful feedback, which greatly enhanced the quality of our manuscript. Your thoughtful comments and suggestions are deeply appreciated.

Round 2

Reviewer 1 Report

Comments and Suggestions for Authors

The authors have improved from the preliminary version and can be accepted in current form

Comments on the Quality of English Language

Appropiate